# Wind Damage and Temperature Effect on Tree Mortality Caused by *Ips typographus* L.: Phase Transition Model

Vladislav Soukhovolsky [1], Anton Kovalev [2], Olga Tarasova [3], Roman Modlinger [4], Zdenka Křenová [5,6], Pavel Mezei [7,8], Jaroslav Škvarenina [9], Jaroslav Rožnovský [10,11], Nataliya Korolyova [4], Andrej Majdák [7] and Rastislav Jakuš [4,7,*]

1 V. N. Sukachev Institute of Forest SB RAS, Akademgorodok 50/28, 660036 Krasnoyarsk, Russia; soukhovolsky@yandex.ru
2 Krasnoyarsk Scientific Center SB RAS, Akademgorodok 50, 660036 Krasnoyarsk, Russia; sunhi.prime@gmail.com
3 Laboratory of Ecosystems Biogeochemistry, Institute of Ecology and Geography, Siberian Federal University, Av. Svobodny 79, 660041 Krasnoyarsk, Russia; olvitarasova2010@yandex.ru
4 ETM, Faculty of Forestry and Wood Sciences, Czech University of Life Sciences, Kamýcká 129, 12900 Prague, Czech Republic; modlinger@fld.czu.cz (R.M.); korolyova@fld.czu.cz (N.K.)
5 Global Change Research Centre AS CR., Bělidla 4a, 60200 Brno, Czech Republic; krenova.z@czechglobe.cz
6 Institute for Environmental Studies, Faculty of Science, Charles University, Benátská 2, 12900 Prague, Czech Republic
7 Institute of Forest Ecology, Slovak Academy of Sciences, Štúrova 2, 96053 Zvolen, Slovakia; mezei@ife.sk (P.M.); majdak@ife.sk (A.M.)
8 W. A. Franke College of Forestry and Conservation, University of Montana, Missoula, MT 59801, USA
9 Faculty of Forestry, Technical University in Zvolen, 96001 Zvolen, Slovakia; jaroslav.skvarenina@tuzvo.sk
10 Institut Celoživotního Vzdělávání, Mendel University in Brno, 61300 Brno, Czech Republic; jaroslav.roznovsky@mendelu.cz
11 Czech Hydrometeorological Institute, Branch Office Brno, 61667 Brno, Czech Republic
* Correspondence: jakus@fld.czu.cz

**Abstract:** The aim of this study was to develop methods for constructing a simple model describing tree mortality caused by *Ips typographus* L. using a minimum number of variables. We developed a model for areas spanning natural mountain forests in the Tatra National Park (Slovakia) and the Šumava National Park (Czech Republic), and in managed Czech forests located in four areas varying in environmental conditions. The model describes the time series of tree mortality dynamics caused by *I. typographus* using two submodels: a long-term dynamics submodel, and a short-term dynamics autoregressive distributed lag(ADL) model incorporating a two year delay and temperature variable averaged over the April-May period. The quality of fit for our models ($R^2$ value) ranged from 0.87 to 0.91. The model was formulated to capture the average monthly temperature effect, a key weather factor. We found that for high-elevation stands located at least 1000 ma.s.l., forest damage was predominantly influenced by May temperatures. For lower-elevation managed forests with warmer climates, the weather effect was insignificant.

**Keywords:** tree mortality; bark beetle; Czech Republic; Slovakia; autoregressive distributed lag model; long-term; short-term

## 1. Introduction

Outbreaks of xylophagous insects, one of the key economically important pests affecting coniferous stands, are typical for forests of Scandinavia, North of the European part of Russia, and mountain forests of Central Europe [1–10]. Over the past decades, extensive bark beetle outbreaks occurred across large areas in Central Europe, Western United States, and Canada causing extensive tree mortality rates and affecting forest ecosystem functioning. Bark beetle outbreaks often follow various natural disturbances such as

fires, windthrow, snowbreaks, drought periods, or following damage by phyllophagous insects [7,11–17].

Multiple studies show that temperature and precipitation fluctuations, which affect both tree physiological processes and insect biological cycles are among the key drivers responsible for outbreak development [8,9,12,18–20]. Windthrown trees also contribute to increasing pest population densities by providing a forage base for bark beetles and altering tree insolation regimes in newly created forest openings [6,21–26]. The reported increase in frequency of bark beetle outbreaks has been attributed to climate warming [4,7,8,10,17,20,27–29]. These factors have been incorporated in various models of bark beetle population dynamics and outbreak risk assessments [3,5,8,30–38]. To date, there are no reliable prediction systems for bark beetle population dynamics. Ďuračiová et al. [39] developed a model for spatial prediction of bark beetle attack; however, a system for the prediction of bark beetle population dynamics, or space- and time-related tree mortality caused by bark beetles, is missing.

Problems arising in the construction of models, in our opinion, are not about the choice of dynamics factors (and these factors are quite well known), but about the choice of functions for describing the influence of these factors on insect dynamics. Issues arising in model design, however, are associated with the choice of functional forms for the best-shaped response, rather than the choice of independent variables themselves, which are well-known. To estimate bark beetle population size, indirect indicators, such as outbreak area [16,40] and tree mortality measured as volume of damaged wood [8–10] are commonly used. Calculations assume that the volume of damaged wood increases proportionally to changes in forest pest population size observed in the current year.

The distribution of *I. typographus* L. follows the distribution of its host tree, Norway spruce (*Piceaabies* (L.) Karsten), which has a continuous range in Scandinavia, Eastern Europe; and Western Russia and a disconnected presence at higher elevations in Central Europe (Alps, Carpathians). Populations of *I. typographus* most often exist at low densities over large areas and periodically build to epidemic population levels that can cause high levels of tree mortality. They forage collectively, either by infesting defenceless hosts in the form of storm-felled or drought-stressed trees, or by overcoming the defences of living trees [41–43].

We hypothesize that bark beetle populations can exist in two phases—the phase of a consistently rarefied state and the phase of an outbreak. In the phase of a consistently rarefied state, population density is extremely low, with individuals colonizing only a small number of available potential host trees. The outbreak occurs when population density exceeds a certain critical threshold and insects feed on all resources available in a given territory [44–46]. However, these works did not propose methods for theoretical calculations of the critical population density. In the present study, an analogue of the first-order phase transition model in physical systems was considered as a theoretical basis for estimating the threshold population density. We used the annual tree mortality measured as volume of damaged wood as an indirect indicator of *I. typographus* population dynamics. Further, we applied the proposed concept to model *I. typographus* L. outbreaks in forests of predominantly Norway spruce that are managed and protected in the Czech Republic and Slovakia. First, we developed a new model (structure of model); then we determined model coefficients (parameters) in different study areas; and later we analysed model performance. Finally, we estimated wind damage and temperature effects on tree mortality caused by bark beetles. We would like to contribute to future considerations, by way of time- and space-related reliable prediction systems for bark beetle population dynamics or tree mortality caused by bark beetles.

## 2. Materials and Methods

### 2.1. Study Area

Our study area incorporated both protected and commercial stands monitored during severe *I. typographus* disturbances (Figure 1). The conservation portion of the study area

was located in state-owned natural montane forests in Tatra National Park (Tatra NP) in Slovakia and Šumava National Park (Šumava NP) in Czech Republic. The part of the study area comprising Czech-managed forests spanned four areas that varied in environmental conditions and tree mortality levels caused by bark beetles (Figure 1, Table 1). In contrast to commercial stands, management methods and sizes of non-intervention zones were subject to change during the study periods. All four areas in managed forests were under the jurisdiction of Military Forests and Farms of Czech Republic, a state-owned forestry company subordinate to the Czech Ministry of Defence.

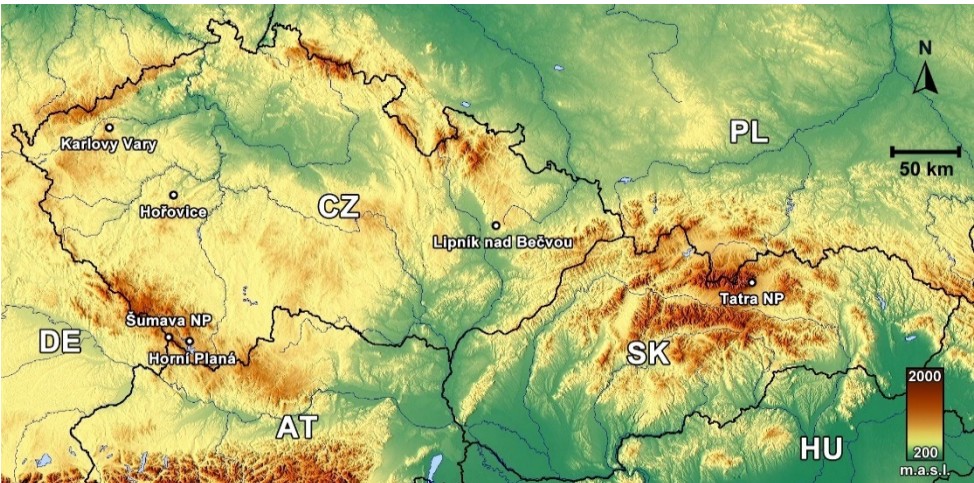

**Figure 1.** Location of study areas (Tatra National Park, Šumava National Park, Karlovy Vary division, Horní Planá division, Lipník nad Bečvou division, Hořovice division); CZ Czech Republic; PL Poland; DE Germany; SK Slovakia; HU Hungary; AT Austria.

**Table 1.** Study area and meteorological station attributes (SA—study area, MS—meteorological station).

| SA Name | SA Coordinates | SA Elevation(m) | SA (ha) | Nearest MS (# id) | MS Coordinates | MS Elevation (m) | Distance between SA and MS (km) |
|---|---|---|---|---|---|---|---|
| Tatra National Park | 49.17° 20.24° | 980–1900 | 2980 | Tatranská Javorina | 49.26° 20.14° | 1013 | MS is located inside of SA |
| Šumava National Park | 48.77° 13.85° | 700–1378 | 68,064 | Churánov (11,457) | 49.07° 13.62° | 1118 | MS is located inside of SA |
| Karlovy Vary division | 50.26° 13.14° | 500–934 | 19,022 | Karlovy Vary airport | 50.20° 12.91° | 603 | 10 |
| Horní Planá division | 48.77° 14.03° | 700–1236 | 19,960 | Černána Podšumaví | 48.74° 14.11° | 740 | MS inside study area |
| Lipník nadBečvou division | 49.63° 17.54° | 500–706 | 27,118 | Červená u Libavé (11,766) | 49.78° 17.54° | 748 | 16 |
| Hořovice division | 49.83° 13.90° | 600–865 | 29,346 | Červená (11,766) | 49.80° 13.75° | 600 | 25 |

### 2.1.1. Tatra National Park

Tatranská Javorina is a protected district of the Tatra NP located on the north-oriented slopes. The deep valleys of study area are dominated by autochthonous mountain Norway spruce.

Populations of *I. typographus* in the study area are largely univoltine, although favourable weather conditions may allow beetles to complete a sister generation [47].

The strategy of non-intervention has not been consistently followed since bark beetle outbreaks erupted at the beginning of the 1990s. Tatra NP pest management approaches were modified several times in the study area, changing from a non-intervention regime prior to 1994, to intensive forest protection management, including application of trap

trees, insecticides, and sanitation cutting in the period of 1995–1996. During 1997–2003, implementation of pest control measures again deviated from non-management approach to sanitary cuttings and intensive application of pheromone trap barriers [48]. From 2004 onward, 1650 ha (55%) of studied forest area gained highest protection level and has not been subjected to any human interventions. Two bark beetle outbreaks occurred in area over the last decades: first was in 1990s, and second after 2004. Regarding case of this study area, data and data acquisition procedure are described in more detail in [4].

### 2.1.2. Šumava National Park

The second non-management study area is in Šumava NP (Czech Republic). The forests, mainly spruce and mixed forests, cover more than 85% (54,439 ha) of the area. Zonal spruce forests (i.e., bog woodlands and mire spruce woods) grow naturally in elevations above 1050 ma.s.l., and occur in mire areas and inversion locations. Large areas of spruce forests have been subjected to significant natural disturbances in last few decades [49,50]. Šumava NP is characterized by a diverse mosaic of old-growth forests, windthrow areas, forests impacted by bark beetles, and areas influenced by past traditional forestry.

Local climate is transitory, with oceanic and continental climate influences throughout year, minute temperature variations, and relatively high rainfall.

Long-lasting debates on zoning and optimal strategies to protect mountain spruce natural stands in this area [51,52] escalated after hurricanes Kyrill and Emma occurred in 2007 and 2008, respectively. Subsequent bark beetle outbreak during 2008–2012 substantially affected NP forests. In January 2007, ~20% of forest area was granted a non-intervention status. Only environmentally sound forest protection activities, such as debarking of lying or standing trees, with all dead wood retained at a site, were allowed on 10% of forest area. In the rest of the Šumava NP territory, to a certain extent, common forest management activities of varying intensity could be implemented. However, chaotic political interventions and management instability eroded this concept [53].

### 2.1.3. Karlovy Vary Division

The Karlovy Vary division is in western Bohemia and includes most of the Doupovské Mountain. The average annual air temperature is 6 °C, and the average annual total precipitation is 600–800 mm [54]. The forest stands are composed of 60% conifers (mainly Norway spruce). Spruce bark beetle has been in the endemic state a long time here. More important is damage by wind, the most serious of which was in 2007 by hurricane Kyrill.

### 2.1.4. Horní Planá Division

Horní Planá division is in the South Bohemian region. Spruce stands dominate in the study area. The stands are managed by the Military Forests and Farms of the Czech Republic. The average annual air temperature is 4–5 °C, and the average annual total precipitation is 800–1000 mm [54].

Tree mortality caused by bark beetles has existed at a low level for a long time. However, situation changed after the hurricane Kyril windstorm in January 2017, which destroyed ten-fold of annual harvest volume overnight, providing an ample breeding base for bark beetle development.

### 2.1.5. Lipník nad Bečvou Division

Lipník nad Bečvou division, another of the Czech army's training polygon since 1946, is in northern Moravia. The average annual air temperature does not exceed 5–6 °C. The average annual total precipitation ranges from 700 to 800 mm [54], but only in the last decade. Wooded area occupies 85% of the division, with spruce-monodominant stands covering ~23,000 ha. A low degree of static stability of stands is responsible for frequent wind disturbances occurring in this area. Furthermore, climate change and wind-induced severe bark beetle outbreak led to a collapse of spruce forests in this area.

### 2.1.6. Hořovice Division

Hořovice division, is a region in central Bohemia, with an average annual air temperature of 5–6 °C, and average annual total precipitation of 600–800 mm [54]. A prevailingly forested Brdy area occupies 95% of the division territory. Forest stands are dominated by Norway spruce covering ~28,000 ha, or about 70% of the area. The bark beetle population has existed in an endemic condition for a long period of time. However, the consequences of hurricane Kyril were similar to those observed in the Horní Planá division. Despite a lower amount of damaged wood (one-fold of total annual harvest volume), the windfall created abundant breeding material for bark beetle proliferation.

### 2.2. Tree Mortality Data

The volume of wind-damaged wood has been recognized as an important driver of spruce ecosystem dynamics because acutely stressed, uprooted, or windfallen trees provide food resources for bark beetle development and population amplification [6,22–26]. As direct assessment of bark beetle population density is complicated, we used a time series of the volume $V(t)$ of bark beetles—induced wood damage (tree mortality) in year $t$ as an indirect indicator of *I. typographus* population dynamics.

Wind and beetle-caused wood damage (tree mortality) data were sourced from field surveys conducted by organizations that manage forests in our study areas in Tatra NP and Šumava NP, as well as from the Military Forest and Farms state enterprise.

### 2.3. Meteorological Data

To ensure the accuracy of weather condition estimates, we used the data obtained by meteorological stations located inside, or in close proximity to, our study area (Table 1). For each study year, daily records of temperature were averaged over monthly periods. To understand the potential effects of climatic conditions on tree mortality, we used mean monthly air temperature.

### 2.4. Modelling Design

We considered outbreaks of forest insects by analogy with first order phase transitions (for example, boiling) in physical systems. For physical systems, first order phase transitions are characterized by a potential function—a function of free energy, which has two local minima $F_1$ and $F_2$ at some states of the object, separated by a local maximum $F_{12}$—a potential barrier. The system can be in one of the states when it has a minimum energy value. During a phase transition under the influence of external factors, for example, changes in the temperature of the environment, the physical system passes from one stable state $X_1$ to another stable state. To define potential function, an approach based on the statistical study of the residence times of the system in different states can be used [55]. To design an outbreak model using this approach, we introduced system state function $F(X)$, which we defined as an inverse value $\frac{1}{f(X)}$ of distribution density function $f(X)$ of population density $X$ values over a long time period, $T$. This function was sufficient and could be realized for all possible states of a system. If probability of a certain state $X_m$ realization for a sufficiently long observation time $T$ is small, then value of function $F(X_m)$ will be large. If state $X_m$ is realized frequently during period $T$, then value of $F(X_m)$ is small.

Function $F(X)$ potentially depends on a large number of various factors and it is difficult to write down dependences of $F(X)$ on these factors. However, $F(X)$ can be classified according to quantity, values, and positions of local minima and maxima. Function $F(X)$ with one global minimum at the value $X = X_1$ will characterize a population with one stable state. Existence of a system near a stable state $X_1$ is associated with implementation of negative feedbacks in the system when its state deviates from value of $X_1$. A state function with two local minima $F(X_1)$ and $F(X_2)$ (where $X_1 \ll X_2$) and one local maximum $F(X_{12})$ between local minima will be characterized as a population with possible transitions from state to state. If the value of $F(X_{12})$ is large, this state is observed very

rarely and system goes from state $X_1$ to $X_2$ and back, skipping over state $F(X_{12})$. Using the approach describing a first-order phase transition model proposed by [56], a state function can be written as an expansion in a Taylor series in some generalized indicator of the system state—order parameter $q$:

$$F(q) = F_0 + \frac{1}{2}a(Z - Z_r)q^2 + bq^4 + cq^6 \tag{1}$$

where $Z$ is an external factor influencing system state; $a$, $b$, $c$ are some constants of expansion.

A general view of the state function $F(X) - F_0$ for a bistable system with two stable states $X_1$ and $X_2$ is shown in Figure 2.

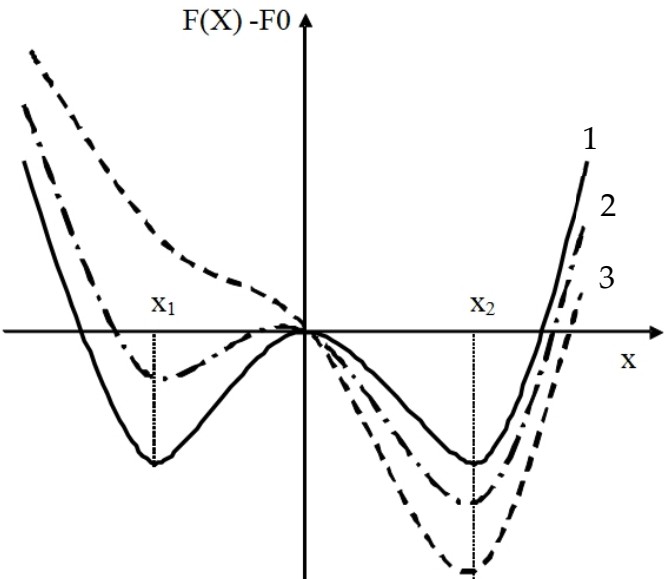

**Figure 2.** Theoretical function $F(X) - F_0$ of the state of a population with two stable states $X_1$ and $X_2$. 1—function without external field; 2—function with weak external field; 3—function with strong external field.

Local minima of a state function can be determined from condition $\frac{dF}{dq} = 0$. After differentiation of Equation (1), we obtained a critical value of $q$, and when this was reached a phase transition occurs:

$$\frac{dF}{dq} = 2a(Z - Z_r)q - 3cq^2 + 4bq^3 = 0 \tag{2}$$

The action of such environmental factors as weather conditions, parasites and predators, amount of available food, within the framework of the first order phase transition model, can be considered as an action of an external field $h$:

$$F(q) = F_0 + \frac{1}{2}a(Z - Z_r)q^2 + bq^4 + cq^6 - qh \tag{3}$$

If external field $h > 0$ increases for a bistable state function (3), then depth of left local maximum will decrease proportionally to the external field (see Figure 2). In this case, at a certain value of the external field, a local minimum at $X_1$ will disappear, system will transfer to a state with a density $X_2$, and an outbreak will occur. A change in sign of the external field and/or an increase in the influence of regulatory factors (for example, an increase in herbivore pressure, and a decrease in volume and quality of available food) will lead to a reverse jump in the state of the population, cessation of the outbreak, and system's return to a stable state.

When analysing fluctuations in state of insect populations, two types of fluctuations can be distinguished: high-frequency and low-frequency. In the case of high-frequency fluctuations, when state of population changes little over time, population will always be near one of stable states $X_1$ or $X_2$. Low-frequency density fluctuations characterize situations when transitions from one stable state to another occur, and characteristic time of such fluctuations—so-called Kramer's time [57]—corresponds to the average time between two adjacent outbreaks.

It is rather difficult to collect data on changes in population density around $X_1$, although probability of a population jump from $X_1$ to $X_2$ state depends on the amplitudes of these fluctuations. Oscillations around $X_2$ value can be measured fairly well due to their large absolute values. In this case, so-called ADL (autoregressive distributed lag) models can be used for modelling [15,46]:

$$X(i) = a_0 + \sum_{j=1}^{k} a_j X(i-j) + \sum_{m=0}^{u} b_m W(i-m) \tag{4}$$

where $X(i)$ is state of pest population in $i$-th year; $a_0 + \sum_{j=1}^{k} a_j q(i-j)$—autoregressive (AR) component of Equation (4), characterizing influence of populations state in $k$ previous years (i.e., we can talk about a lag in time series of population), $W(j)$ are external factors (weather conditions, food supply) in $j$-th year; $k$, $u$, $a_j$, and $b_m$ are constants, $k$ is order of autoregression and characterizes period of past seasons influence on current population state, $n$ is period during which external factors affect current state of population.

We used following data fitting quality indicators with model (6) [15]:

1. A determination coefficient $R^2$, characterizing proportion of variance explained by model. The adjusted determination coefficient (adj.$R^2$) is used to account for influence of variables number. In this case, variables number for different models is the same, so $R^2$ is shifted in relation to adj.$R^2$ by the same value.
2. $t$-test for estimation difference of model coefficients from zero.
3. F-test to test hypothesis that all coefficients of linear model are equal to zero.

Coefficients $a_j = \frac{\partial q(t)}{\partial q(t-j)}$ characterize sensitivity of current population state to changes in previous seasons; coefficients $b_j = \frac{\partial q(t)}{\partial W(t-j)}$ characterize sensitivity of current population state to the influence of external factors.

In Equation (4), following variable values are known: on the left side of equation, time series $\{X(i)\}$ of system state dynamics, and on the right side—series $\{X(i-j)$ (in fact, the same time series of "forest-insects" state taken with some delay $j$), and a number of indicators $W(i)$ selected from minimum discrepancies condition of model series to the time series $\{X(i)\}$ of population count data. In fact, this means that Equation (4) should be considered as a linear regression equation, for which estimation of orders of $k$ and $u$ on the right side are needed; and then, using standard procedure for finding coefficients of linear regression equations, calculate coefficients $a_j$ and $b_m$.

It is possible to use methods of autocorrelation analysis [58–61] and calculate partial autocorrelation function (PACF) to estimate order $k$ of the AR-component of model. However, such a calculation is performed correctly only if the studied time series is stationary and standard deviations do not change over time [58,62]. If these conditions are not met, and there is a trend in population density values and (or) a time-varying range of density fluctuations, then it is necessary to transform the observed time series without losing the properties of the series of interest. Hence, it could be transformed into a Linear Time Invariant (*LTI*) series. A *LTI* series should have a time-constant mean, a time-constant mean variance, and a time-constant oscillation frequency of variable. To assess the quality of the model, four criteria were used: value of determination coefficient $R^2$ should be close to 1, values of regression coefficients in Equation (4) should be significant according to $t$-test, transformed series of accounting data, and model series should be synchronous

(synchronicity is assessed by characteristics of cross-correlation functions between these series). We can say that it adequately described the dynamics of the studied population if the model meets all these criteria.

### 2.5. Data Processing

To construct ADL models to reduce variance of population state values, we transformed count data to a logarithmic scale of densities. Low-frequency filtering allowed us to identify long-term trends in densities, and high-frequency filtering was used to reduce possible counting errors. To isolate the high-frequency component, i.e., to select oscillations with higher frequencies, we applied a Hann filter [63] in the following functional form: $f_0 = \frac{1}{4 \cdot \tau}$, where $\tau = 1$ is the time between two adjacent counts of population state, represented by a formula with weighting coefficients 0.24, 0.52, and 0.24 for values $\ln (X - 1)$, $\ln X(t)$, and $\ln X(t + 1)$, respectively:

$$\ln \hat{X}(t) = 0.24 \ln X(t - 1) + 0.52 \ln X(t) + 0.24 \ln X(t + 1) \tag{5}$$

Thus, to model dynamics of population using data from long-term records of tree mortality (wood damage), researcher can use, on the one hand, first-order phase transition models to explain outbreak phenomenon itself and, on the other hand, ADL-models to describe changes in the population state during an outbreak. Parameters of ADL model can be calculated using statistical package Statistica 10.

### 3. Results

#### 3.1. Model and Model Coefficients

As it is rather difficult to estimate population density for xylophagous populations, an indirect indicator is usually used—area $S$ or volume $V$ of bark beetle attacked wood in one year. In this case, to construct the state function $F(\ln V)$ instead of $F(X)$, where $F(\ln V) = \frac{1}{f(\ln V)}$ and $f(\ln V)$—distribution density function of values $\ln V$. For a long time period $T_1$, outbreaks are not observed, volume of wood attacked by bark beetle is small, probability $f(V_1) \propto \frac{T_1}{T}$ (were $T$—general time of stand observation) that stand is in an inter-outbreak state is high, and then value $F(\ln V_1) = \frac{1}{f(\ln V_1)}$ is small. Because the transition from a state with a low $V_1$ value to a state with a high tree mortality caused by bark beetle (volume of wood attacked by bark beetle), level $V_2$ occurs quickly, and $F(\ln V_{12})$ value for this transition state $V_{12}$ is large.

Using data on the volume of wood attacked by bark beetles, we constructed a state function of "forest—insects" system as an inverse function of density distribution of tree mortality volume by caused bark beetle $f(\ln V)$ over a long observation period $T$ (Figure 3).

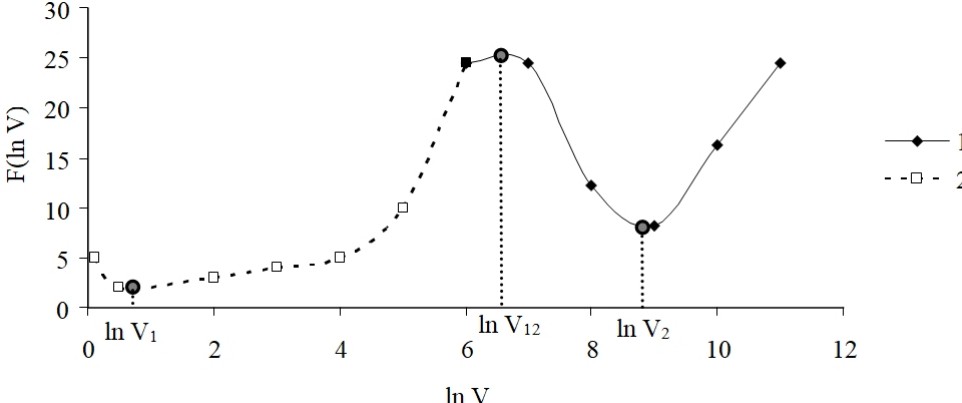

**Figure 3.** State function $F(\ln V)$ (1) for volume of wood attacked by bark beetles at Karlovy Vary division study area and a hypothetical function $Fr(\ln V)$ (2) for low density state.

As seen in Figure 3, outbreak phase can be characterized by value of ln $V_2$, a potential barrier is characterized by value of ln $V_{12}$, and phase of rarefied state can be characterized by value of ln $V_1$.

The partial autocorrelation function (PACF) was used as an indicator for connection between ln $V(t)$ and ln $V(t-k)$. The PACF calculation is correct only for stationary time series, therefore, to start calculations at Karlovy Vary division study area, we selected linear trend ln $VTR(t) = A - Bt$ of series {ln $V(t)$} and then will calculate PACF for time series {{$\Delta$ ln $V(t)$} = {ln $V(t)$ − ln $VTR(t)$}} (Figure 4).

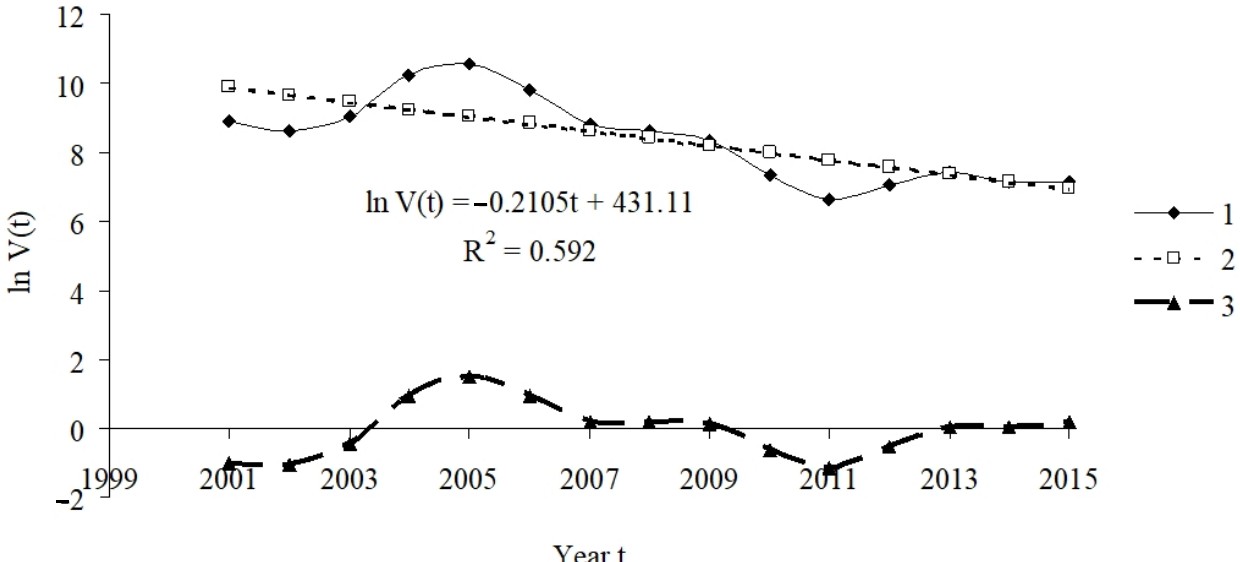

**Figure 4.** Time series (1), trend (2) of wood attacked by bark beetle at Karlovy Vary division study area and stationary component (3) of time series.

The PACF of stationary component (3) of time series in Figure 3 for wood attacked by bark beetle at Karlovy Vary division is shown in Figure 5.

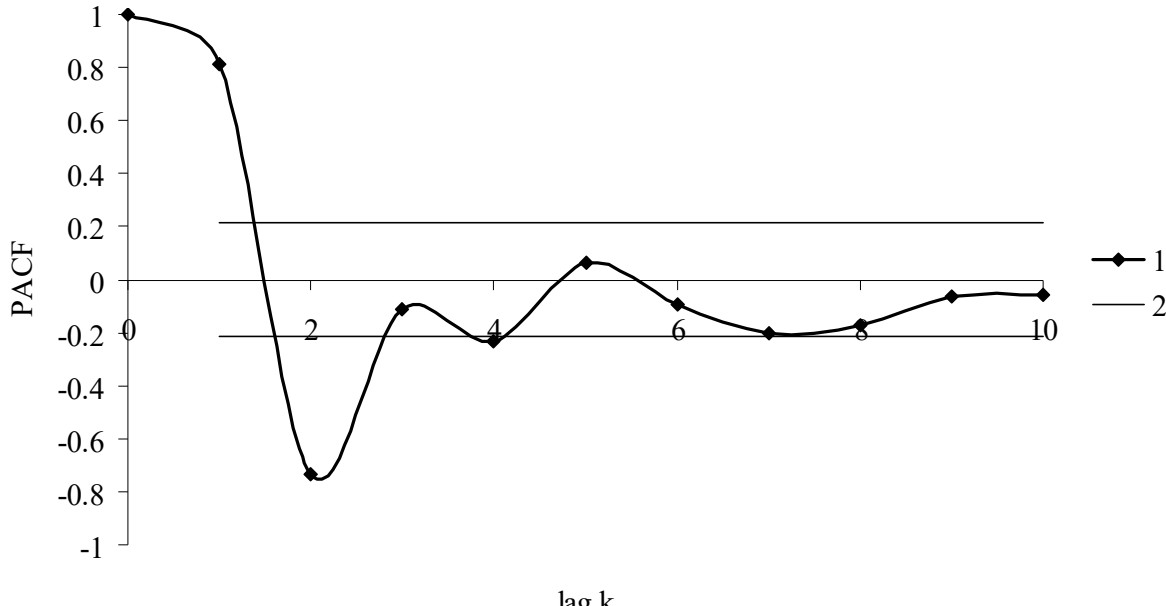

**Figure 5.** Partial autocorrelation function (PACF) of the stationary component of time series of wood attacked by bark beetles at the Karlovy Vary division study area. In the legend, 1—is PACF, 2—is 90% standard error of PACF.

As follows from Figure 5, the order of autoregression k = 2. Current fluctuations in the system state are not random and depend on two previous seasons. To assess the influence of external factors (weather or volume of windfallen trees) on the system state, it is difficult to use a model of phase transitions and autoregression. Therefore, to assess effect of weather, we will choose values with maximum coefficient of determination $R^2$ for ADL model. Under these conditions, ADL model for first differences $\Delta \ln V(t)$ will have the following form:

$$\Delta \ln V(t) = a_0 + a_1 \cdot \Delta \ln V(t-1) + a_2 \Delta \ln V(t-2) + \sum_{m=0}^{u} b_m \ln W(t-m) \quad (6)$$

In this case, the maximum value of $R^2$ is achieved at the following values of coefficients of ADL model for investigated plot (Table 2).

**Table 2.** Coefficients of ADL model and their significance for Karlovy Vary division (Variable abbreviations are as follows: *T*—average monthly temperature, in May for year t, *W*—volume of damaged wood by wind in year $(t-1)$). $\{\Delta \ln V(t)\} = \{\ln V(t) - \ln VTR(t)\}$—detrended part of wood volume damaged by bark beetle.

| Variables | Coefficients | Std.Err. | *t*-Test | *p*-Value |
|---|---|---|---|---|
| $a_0$ | 2.212 | 2.713 | 0.815 | 0.438 |
| $T(Mai, t)$ | 0.231 | 0.097 | 2.386 | 0.044 |
| $\ln W(t-1)$ | −0.382 | 0.316 | −1.208 | 0.262 |
| $\Delta \ln V(t-2)$ | −0.475 | 0.266 | −1.786 | 0.112 |
| $\Delta \ln V(t-1)$ | 1.344 | 0.224 | 5.987 | 0.000 |
| $R^2$ | | 0.910 | | |
| $adj.R^2$ | | 0.87 | | |
| *F-test* | | 19.350 | | |

Dynamics of tree mortality by *I. typographus* at the Karlovy Vary division study area and model calculations are shown in Figure 6.

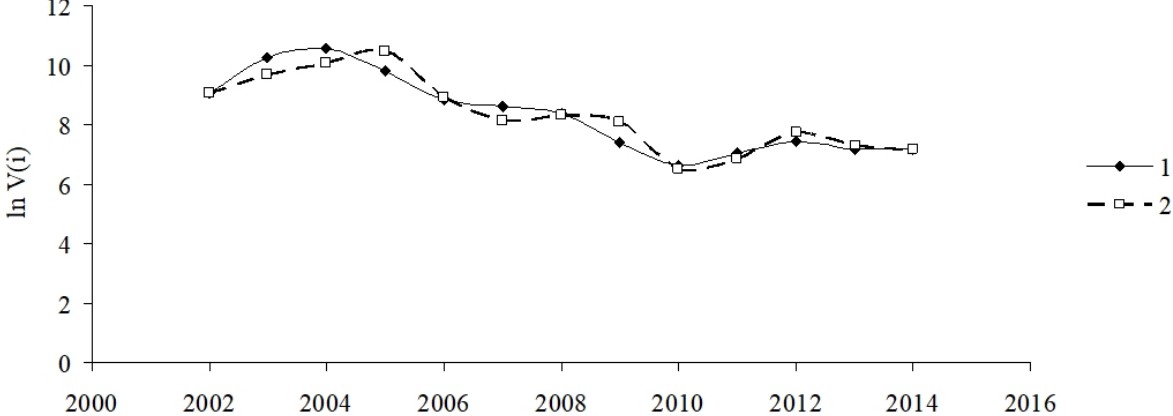

**Figure 6.** Empirical (1) and modelled (2) time series of wood damage caused by bark beetle in the Karlovy Vary division.

The synchronism of the empirical and modelled time series of wood damage caused by bark beetle is estimated using a cross-correlation function (CCF) as shown for Karlovy Vary division study area (Figure 7).

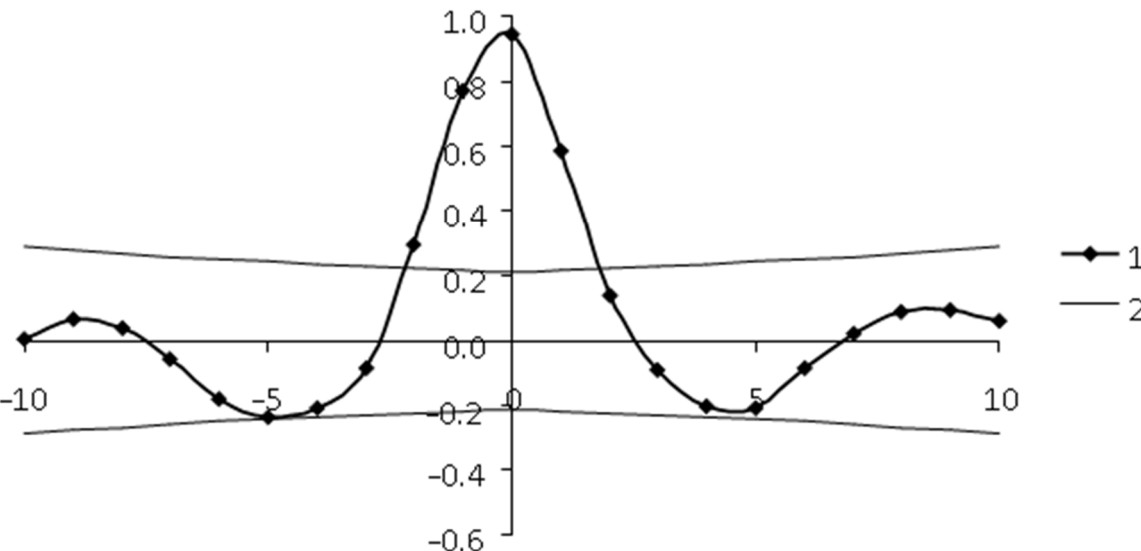

**Figure 7.** Cross-correlation function (CCF) (1) of natural and model time series of tree mortality caused by bark beetle at the Karlovy Vary division study area and standard error of CCF (2).

Similar calculations and figures for the stationary component of time series, the partial cross-correlation function, the empirical and modelled time series of tree mortality caused by bark beetle, the cross-correlation function of empirical and model time series of damaged forest, were carried out for other study areas. Those data are available both on the intensity of wind damage and monthly weather conditions. The quantified values of ADL model parameters, which considered the effect of windthrows and air temperature in certain months for other study areas, are shown in Table 3.

**Table 3.** Model parameters to estimate the volume of wood damaged by bark beetle (tree mortality) during an outbreak in different study areas (for abbreviations refer to Table 2).

| Variable | Coefficient | Std. Error | *t*-Test | *p*-Value |
|---|---|---|---|---|
| | | Tatra National Park | | |
| $a_0$ | −1.708 | 0.765 | −2.233 | 0.039 |
| $T(Mai,t)$ | 0.120 | 0.056 | 2.144 | 0.047 |
| $\ln W(t)$ | 0.045 | 0.035 | 1.279 | 0.218 |
| $\Delta \ln V(t-2)$ | −0.801 | 0.128 | −6.262 | <0.001 |
| $\Delta \ln V(t-1)$ | 1.533 | 0.131 | 11.713 | 0.000 |
| $R^2$ | | 0.911 | | |
| *adj.*$R^2$ | | 0.88 | | |
| *F-test* | | 43.66 | | |
| | | Šumava National Park | | |
| $a_0$ | 3.367 | 1.031 | 3.267 | 0.003 |
| $T(Mai, t-2)$ | 0.313 | 0.162 | 1.935 | 0.065 |
| $\ln W(t-3)$ | 0.305 | 0.123 | 2.469 | 0.021 |
| $\Delta \ln V(t-2)$ | −1.242 | 0.219 | −5.660 | <0.001 |
| $\Delta \ln V(t-1)$ | 1.599 | 0.164 | 9.722 | <0.001 |
| $R^2$ | | 0.88 | | |
| *adj.*$R^2$ | | 0.84 | | |
| *F-test* | | 40.87 | | |

**Table 3.** *Cont.*

| Variable | Coefficient | Std. Error | *t*-Test | *p*-Value |
|---|---|---|---|---|
| | | Horní Planá division | | |
| $a_0$ | −3.019 | 2.424 | −1.245 | 0.248 |
| $T(Mai, t)$ | 0.192 | 0.116 | 1.650 | 0.138 |
| $\ln W(t)$ | 0.386 | 0.152 | 2.545 | 0.034 |
| $\ln V(t-2)$ | −0.409 | 0.152 | −2.696 | 0.027 |
| $\ln V(t-1)$ | 1.109 | 0.192 | 5.788 | 0.000 |
| $R^2$ | | 0.872 | | |
| $adj.R^2$ | | 0.78 | | |
| *F-test* | | 13.580 | | |
| | | Lipník nad Bečvou | | |
| $a_0$ | −6.086 | 1.460 | −4.170 | 0.004 |
| $T(April, t)$ | 0.047 | 0.037 | 1.277 | 0.242 |
| $\ln W(t)$ | 0.478 | 0.117 | 4.105 | 0.005 |
| $\Delta \ln V(t-2)$ | −0.717 | 0.124 | −5.757 | 0.001 |
| $\Delta \ln V(t-1)$ | 0.384 | 0.143 | 2.676 | 0.032 |
| $R^2$ | | 0.911 | | |
| $adj.R^2$ | | 0.87 | | |
| *F-test* | | 17.84 | | |
| | | Hořovice division | | |
| $a_0$ | −3.047 | 0.933 | −3.266 | 0.014 |
| $T(Mai, t)$ | 0.048 | 0.042 | 1.146 | 0.289 |
| $\ln W(t)$ | 0.230 | 0.082 | 2.806 | 0.026 |
| $\ln V(t-2)$ | −0.373 | 0.189 | −1.968 | 0.090 |
| $\ln V(t-1)$ | 0.979 | 0.171 | 5.730 | 0.0007 |
| $R^2$ | | 0.88 | | |
| $adj.R^2$ | | 0.79 | | |
| *F-test* | | 12.25 | | |

Parameter $V$ in $t-2$ proved to be significant in nearly every study area, except the Karlovy Vary division study area. Damage in the previous year was significant in every study area.

A series of field data on the dynamics of tree mortality caused by bark beetle is shown for the SA Tatra NP, Šumava NP, Horní Planá, Lipník nad Bečvou, and Hořovice divisions (Figure 8).

Two outbreaks were observed in the studied period in Tatra NP. The first outbreak occurred in the mid-1990s, and the second outbreak was observed after 2004. The tree mortality remained relatively high for a long period of time. Such outbreaks in a period of less than 10 years are unusual for higher elevation forests. In Šumava NP, tree mortality caused by bark beetles remained relatively unchanged during the study period, although it was maintained at high levels.

The data for Horní Planá resembles a typical epidemic curve, with low fluctuation of tree mortality.

Tree mortality in the Lipník nad Bečvou study area resembles a typical permanent high level of tree mortality. The mature spruce stands were destroyed by the year 2018.

Tree mortality in SA Hořovice was much more stable.

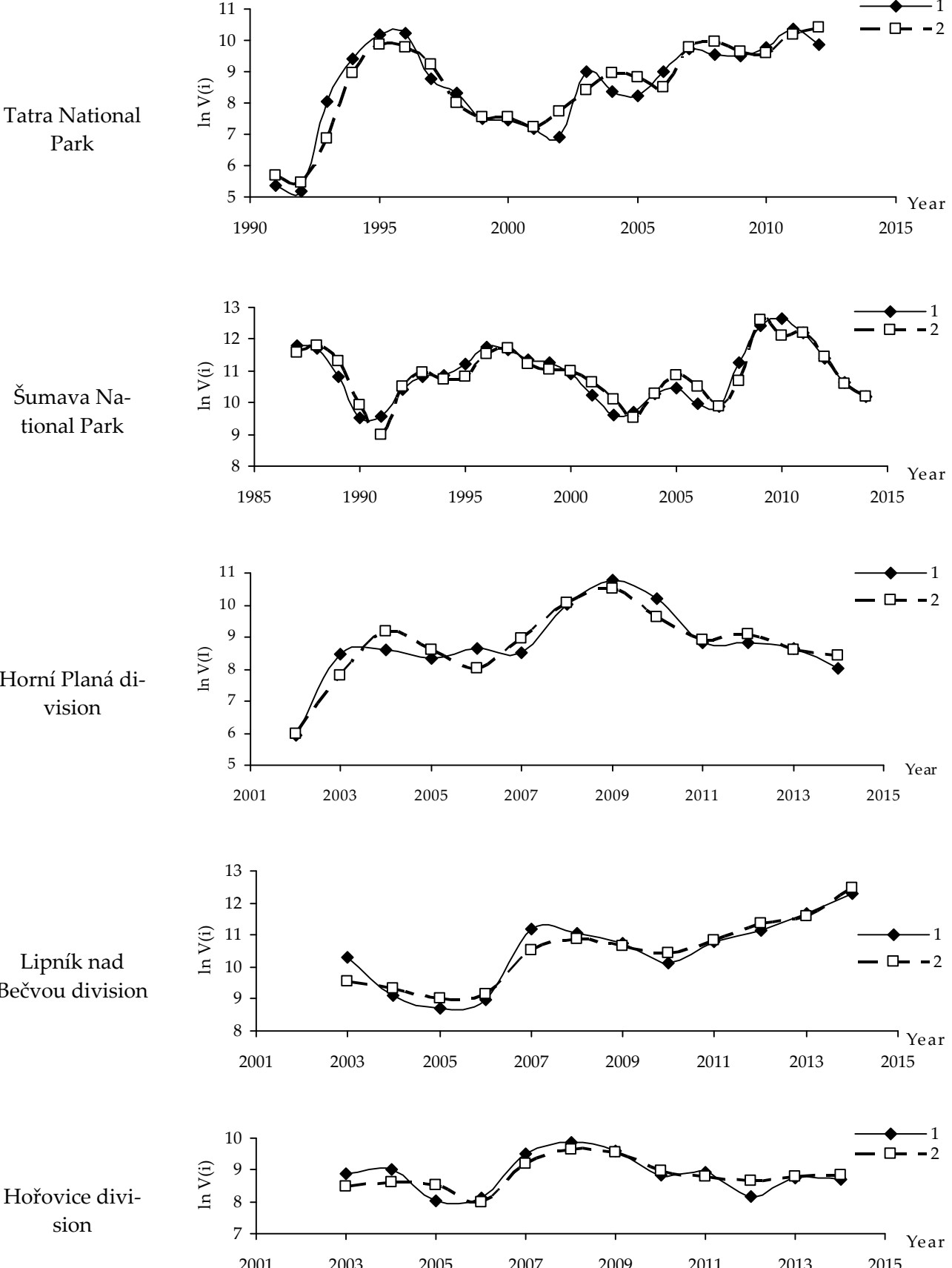

**Figure 8.** Field (1) and modelled (2) levels of wood attacked by bark beetle at the sample plots in the different study areas.

*3.2. Model Performance*

According to calculation of PACF, order of autoregression for model of forest attacked by bark beetle populations is equal to 2 (Table 3). That is, current level of tree mortality caused by bark beetle is determined by levels of tree mortality caused by bark beetle in previous two years, and a positive feedback is observed between tree mortality caused by bark beetle in past year and current year: higher level of tree mortality caused by bark beetle in past year causes a higher level in current year. On the contrary, there is a negative feedback between tree mortality caused by bark beetle in year before last and current year—higher level of tree mortality caused by bark beetle in year $(t-2)$, lower level is in year t. These relationships can be explained by increasing population density at a high level of tree mortality caused by bark beetle per year $(t-1)$ that enhances food resources available for next beetle generation, intensifying mass attacks on host trees. On the contrary, if food supply decreases during two seasons, trees damaged in year $(t-2)$ become unsuitable for colonization in year $t$. Model (6) indicates that an important driver of beetle-induced tree mortality dynamics is the volume of wind-fallen trees. Windfalls commonly occur in winter and mostly depend on tree growing conditions. Trees felled in the current year become a food resource for bark beetles in the following year. Weather conditions affect the dynamics of damage in different ways. For example, in the Šumava NP trial plot, it was found that current level of tree mortality caused by bark beetle significantly depends on average May temperature of year $(t-2)$. Sign of damage level from weather changes sensitivity coefficient is positive. It indicates that warmer weather in May is associated with a higher level of tree mortality caused by bark beetle two years later. The lag between temperature rise and level of tree mortality caused by bark beetle can be related to the physiological response of trees, manifested in decline of its resistance capacities after wind damage. In Karlovy Vary study area and Tatra NP study area, average monthly temperature in May in year $t$ was significant. In remaining managed study areas at lower elevations, monthly average temperatures were not statistically significant.

The question arises: under what conditions will outbreak stop? From perspective of proposed approach, to stop an outbreak, it is necessary to reduce effects of external field to an extent that a minimum of local fluctuations in the current level of tree mortality caused by bark beetle is reached. In other words, outbreak ceases under impact of decline in bark beetle population density. Since model (6) is a second-order autoregression model (Yule model), spectral density $p(f)$ of plant damage time series according to model (6) will be described by Equation (7) [13,64,65]:

$$p(f) = \frac{2\sigma_\varepsilon^2}{1 + a_1^2 + a_2^2 - 2a_1(1 - a_2)cos2\pi f - 2a_2 cos4\pi f} \tag{7}$$

Empirical and theoretical spectral density of tree mortality time series caused by bark beetle calculated according to Equation (7) are shown in Figure 9.

Frequencies $f_{max}$ = 0.142, which correspond to the maximum spectral density of time series of 3.64 for tree mortality caused by bark beetle, are close to the empirical and theoretical spectra and wave length $L = \frac{1}{f_{max}} = \frac{1}{0.142} \approx 7 \; years$ as indicated in Figure 9.

As can be seen from Equation (7), the cyclicity of outbreaks depends on the values of parameters of ADL model. If the value of $a_1$ (susceptibility of $\frac{\partial \Delta \ln V(t)}{\partial \Delta \ln V(t-1)}$ volume of wood attacked by bark beetle in year $t$ to the volume of wood attacked by bark beetle in year $(t-1)$) increases, then the frequency of the spectrum maximum will shift towards lower frequencies, and the maximum of the spectral power will increase (Figure 10, curve 2). If, on the contrary, the value of $a_1$ decreases, then the frequency of the spectrum maximum will increase, and the value of the maximum spectral power will decrease (Figure 10, curve 3).

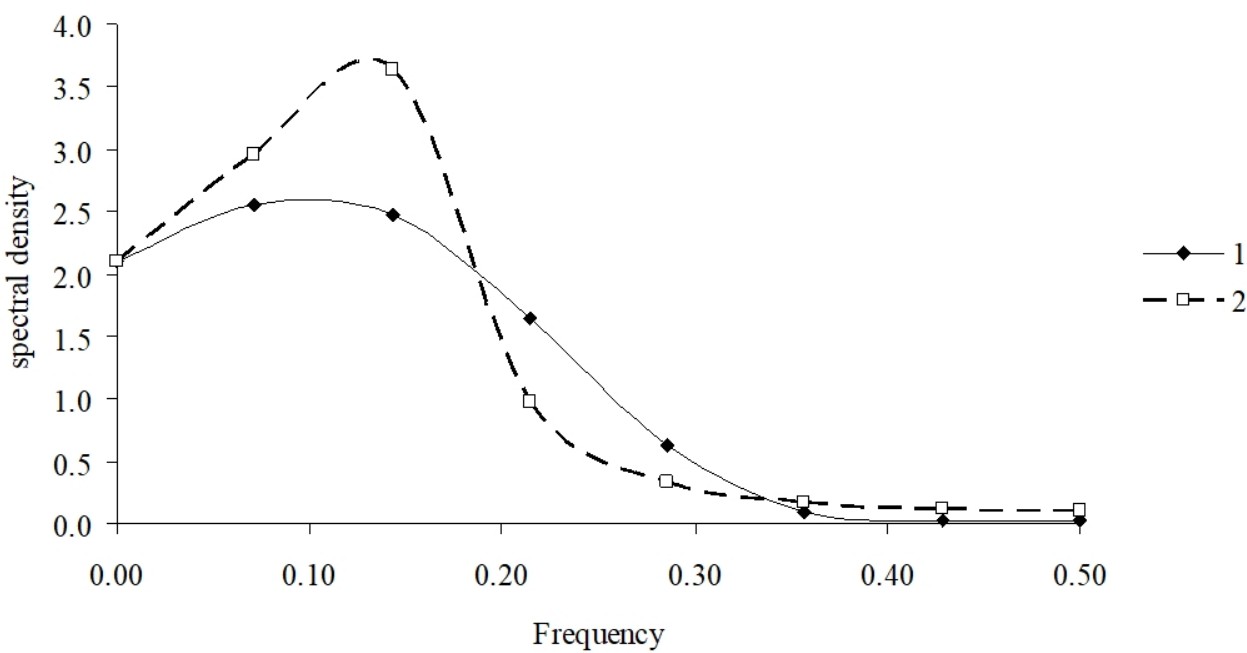

**Figure 9.** Empirical (1) and theoretical (2) spectral density of tree mortality time series caused by bark beetle in the Karlovy Vary division study area.

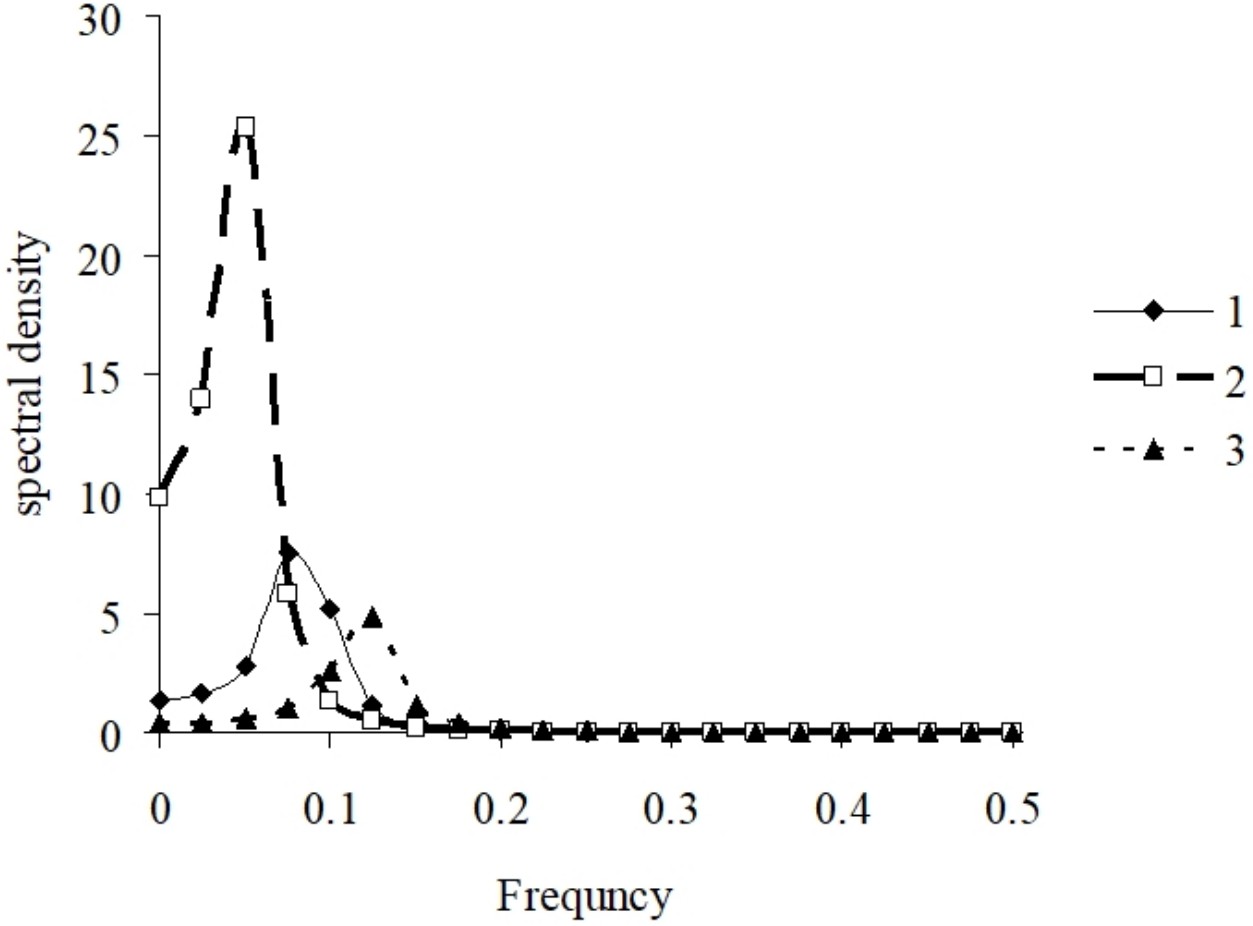

**Figure 10.** Theoretical spectral density of time series of tree mortality caused by bark beetle. Curve 1 ($a_1$ =1.533, $a_2$ = −0.801) (ADL model for the Tatra National Park study area); curve 2 ($a_1$ = 1.7, $a_2$ = −0.801); curve 3 ($a_1$ = 1.3, $a_2$ = −0.801).

### 3.3. Estimation of Wind Damage and Temperature Effect on Tree Mortality Caused by Bark Beetle

How strongly do such factors as the volume $W(t)$ of trees felled by the windfall and the temperature $T(t)$ of the environment influence the dynamics of tree mortality caused by bark beetle? If we assume that there are no wind impacts on the stands and $W(t) = 0$, and changes in air temperature do not affect fluctuations in the dynamics of bark beetle population and level of connected tree mortality, then the stability of the tree mortality process caused by bark beetle will depend on the $a_1$ and $a_2$ components of model (6) [64]:

$$
\begin{cases}
a_1 + a_2 < 1 \\
a_1 - a_2 > -1 \\
-1 < a_2 < 1
\end{cases}
\tag{8}
$$

Using data in Tables 2 and 3, we estimated realization of inequalities (Equation (8)) (Table 4).

**Table 4.** Values of inequalities (Equation (8)) and stability margin for series of studied tree mortality caused by bark beetle $\{\Delta \ln V(t)\}$ at the different survey areas.

| Parameters and Conditions | Study Area | | | | | |
|---|---|---|---|---|---|---|
| | Karlovy Vary Division | Tatra NP | Šumava NP | Horní Planá Division | Lipník nadBečvou Division | Hořovice Division |
| $a_2$ | −0.48 | −0.80 | −1.24 | −0.41 | −0.72 | −0.375 |
| $a_1$ | 1.34 | 1.53 | 1.60 | 1.11 | 0.38 | 0.98 |
| $a_1 + a_2 < 1$ | 0.26 < 1 | 0.73 < 1 | 0.36 < 1 | 0.70 < 1 | −0.33 < 1 | 0.61 < 1 |
| $a_1 - a_2 > -1$ | 1.82 > −1 | 2.33 > −1 | 2.84 > −1 | 1.52 > −1 | 1.10 > −1 | 1.35 > −1 |
| $-1 < a_2 < 1$ | $\|a_2\| < 1$ | $\|a_2\| < 1$ | $\|a_2\| > 1$ | $\|a_2\| < 1$ | $\|a_2\| < 1$ | $\|a_2\| < 1$ |
| $\eta$ | 0.13 | 0.10 | 0.17 | 0.29 | 0.28 | 0.37 |

In accordance with Table 4, for all study areas, except for Šumava, conditions (8) are satisfied. That is, $a_1$ and $a_2$ components in Equation (6) can be considered stable. Table 4 shows values of stability margin for different study areas. Also from Table 4, values of stability margin are small, and in general it can be concluded that in absence of a windthrow and elevated air temperatures, stability of tree mortality volumes caused by bark beetle and, therefore, dynamics of bark beetle population is rather small, and a loss of stability for bark beetle population dynamics is possible.

In addition to conditions (8), for AR models of forest insects population dynamics, there are important characteristics that make it possible to assess changes in population state under various external environment transformations, and proper population characteristics, such as stability margin $\eta$ [66]. Stability margin characterizes proximity of this point to the boundaries of stability zone. For discrete systems, Mikhailov criterion and Mikhailov hodograph are used [67] to estimate the stability margin.

To assess stability of $a_1$ and $a_2$ components in Equation (6), it is necessary to write down the characteristic equation $D(z) = z^2 - a_1 z + a_2$, replace variable $z$: $z = \frac{1+\nu}{1-\nu}$ and build Mikhailov hodograph $D(j\nu) = D(z)\,|_{z = \exp(j\nu)}$. (where $j = \sqrt{-1}$). To construct the Mikhailov hodograph, the real and imaginary parts of polynomial are separated $D(j\nu)$. The stability margin $\eta$ is the circle radius centred at the point $z = 0$, which can be written inside Mikhailov hodograph [68].

By definition, value of margin stability $\eta \geq 0$ and smaller value of $\eta$, provide a greater likelihood of a "breakdown" and a loss of system stability under external influences. Despite seemingly complicated calculation procedure, stability margin is estimated using a simple program in MATLAB package [68], and only values of coefficients of AR components of Equation (6) are required for calculations.

## 4. Discussion

### 4.1. Model

Tree mortality by bark beetle can be considered as transitions between two stable (or metastable) states in the bistable system of the pest population: a state with a low-density level $X_1$ (and therefore a low level of damage $V$ by the pest), and a state with a high level of population density $X_2$ (and therefore with a high level of tree mortality). In this case, the level of forest damage by the windfall plays the role of the external field [46]. The proposed model makes it possible to identify key factors in the dynamics of insect infestation in forests—the current values of damage after a windfall, average temperature of April or May, tree mortality caused by bark beetle in the previous two years. The proposed model of tree mortality does not require knowledge of bark beetle populations and tree state characteristics. Forest damage by wind, tree mortality by bark beetles and weather data are sufficient to build a model.

Further research is needed to model parameters of local tree mortality. A calculation showed that tree mortality by a bark beetle is not random and depends on both the history of past tree mortality and the current weather. However, using the proposed model for a short-term forecast of tree mortality dynamics is difficult and, in this case, it is necessary to predict future weather in a crucial seasonal time period for *Ips typographus* [34]. However, according to the data of the past tree mortality and the weather conditions in April–May, the risk of damage can be estimated by the order of autoregression $k$ depending on the values of derivatives and maximum potential of a bistable system [46,58]. It should be noted that the proposed ADL models do not describe a long-term state with a low population density due to the lack of field data for these periods. To construct a general model of bark beetle population dynamics (or dynamics of forest tree mortality associated with it), data are needed both in the state of outbreak and in a stable state with a low level of density. In this case, it is possible to construct a more general model of population dynamics (and tree mortality) as analogues of a first-order phase transition model in physical systems with two stable states and oscillations around each of the states.

### 4.2. Model Coefficients: Environmental and Internal Factors

The coefficients of the model differ in the key weather factor: average monthly temperature. These differences may be due to differences in landscape characteristics and elevation of the study areas. For forest stands at high elevation (not lower than 1000 ma.s.l.) (Tatra National Park study area, Šumava National Park study area, and Horní Planá division), damage is influenced by the weather in May. As shown at Table 3 the *t*-test value and the *p*-level coefficients at variable $T$ are significant for $p \leq 0.047$ (Tatra National Park), $p = 0.065$ (Šumava National Park), $p = 0.138$ (Horní Planá division). For lower-lying forests in the Lipník nad Bečvou division study area (500–650 m a.s.l.) and in the Hořovice division study area (600–800 m a.s.l.), with warmer climates, the model coefficients for weather are insignificant (see *t*-test values and *p*-level with variable T in Table 3 for Lipník nad Bečvou and Hořovice divisions), that is, for these territories the damage intensity does not depend on the weather conditions. It should be noted that average temperatures of spring months for the test areas of Tatra National Park, Šumava NP, and Horní Planá division study area, are from 4.8 to 5.1 °C. For Lipník nad Bečvou division study area and Hořovice division study area at lower elevations, average temperatures for spring months were approximately 0.5 °C higher and ranged from 5.4 to 5.5 °C.

Comparing coefficients for variable $\ln V(i-1)$, which characterize susceptibility $\frac{\partial \ln V(i)}{\partial \ln V(i-1)}$ of current tree mortality caused by bark beetle $\ln V(i)$ to the previous year's damage $\ln V(i-1)$, it can be said that in forest stands at high elevations (approximately 1000 ma.s.l.), values of these coefficients were maximum, whereas for Lipník nad Bečvou study area (elevation approximately 500 ma.s.l.) values of these coefficients were minimal. Differences in population dynamics of *I. typographus* related to the different elevation of study areas were described by [69] for areas of Slovak Republic and by [70] for areas of Czech Republic. Lipník nad Bečvou study area is located at an especially (relatively)

low elevation. Spruce forests are at lower elevations in Czech Republic and are seriously affected by drought, especially in the broader region surrounding area of this model [17,71]. Under acute or chronic drought stress the defence abilities of spruce are minimal [72].

The coefficients of the variable ln *W* for all plots, except for Tatra National Park, are approximately equal. The small value of this coefficient for the Tatra National Park study area indicates that the volume of trees which fell under windthrow did not affect tree mortality caused by bark beetles. This finding could possibly be explained by the dominance of high elevations and steep northern-oriented slopes in the study area. Such conditions are not favourable for colonization of downed trees by *Ips typographus* [73].

## 5. Conclusions

Models that account for autoregressive properties of population densities are well known [74–77]. However, the combination of phase transition models and ADL-models make it possible to describe different stages of forest insect's outbreaks. According to the first-order phase transition model, outbreak occurs because of system transition from stable state with low density $X_1$ to a state of high density $X_2$. An outbreak occurs either with fluctuations around $X_1$ (this is possible if height of the potential barrier between states $X_1$ and $X_2$ is small), or under the influence of an external field, when state $X_1$ disappears. For stands damaged by *Ips typographus*, wood volume felled by wind acts as an external "field". In the absence of wind, the damage level fluctuates around $X_1$ value. After exposure to an external "field" and a jump into the outbreak state with a volume of damaged wood near $X_2$, level of damaged wood fluctuates around $X_2$ (with effect of weather and windthrow) according to the proposed ADL-model.

Phase transition models of the first kind and an ADL-model were previously used to model population dynamics and outbreaks of phyllophagous insects [46]. For these models, pest population densities were used as a variable. Variables such as the density of parasites or predators (which are extremely difficult to account in the field) were not used for modelling and were replaced by negative feedback from previous density of simulated population.

The modelled system of a stand damaged by bark beetle differs from such classical systems as "predator-prey" or "resource-consumer" (where a population of bark beetle is considered as a consumer, and trees as an available resource). The system is described by only one variable—volume of damaged wood, and there is no data about bark beetle density. The concept of an outbreak as a first-order phase transition and an ADL-model of simulated variable oscillations (dead wood volumes) near the metastable state $X_2$ allows us to solve the problem of a lack of field measurement data.

Calculations performed within the framework of the proposed model have shown that the use of a fairly small amount of data from past seasons and the meteorological parameters make it possible to predict current forest damage caused by insects. As follows from the values of determination coefficients $R^2$, the ADL model accountedfor at least 87% variance in the dynamics of forest stand damage. The proposed model has the potential to improve forecasting bark beetle population systems.

**Author Contributions:** Conceptualization, V.S., O.T. and R.J.; methodology, V.S., O.T. and R.J.; software, A.K.; data curation, V.S., O.T., A.K., P.M., R.M., Z.K, J.Š., J.R. and R.J.; writing—original draft preparation, V.S.; writing—review and editing, V.S., O.T., A.K., P.M., R.M., Z.K., J.Š., J.R., A.M., N.K. and R.J.; visualization, V.S., A.K. and A.M. All authors have read and agreed to the published version of the manuscript.

**Funding:** Contribution of V.Soukhovolsky and O.Tarasova in article supported by Russian Scientific Foundation (research project number 22-24-00148). Contribution of other authors was supported by grants: "EXTEMIT-K", No. CZ.02.1.01/0.0/0.0/15_003/0000433 financed by OP RDE and by the Slovak Research and Development Agency APVV–15–0761, APVV-18-0347 and by grant No. QK1910480, "Development of integrated modern and innovative diagnostic and protection methods of spruce stands with the use of semiochemicals and methods of molecular biology", financed by the Ministry of Agriculture of the Czech Republic; "Factors of Norway spruce survival during large-scale

disturbance in NP Šumava" No. A_21_09 funded by the Internal Grant Agency FFWS CULS in Prague and The Ministry of Education, Youth and Sports of CR within the CzeCOS program, grant number LM2018123.

**Institutional Review Board Statement:** Not applicable.

**Informed Consent Statement:** Not applicable.

**Data Availability Statement:** The data presented in this study are available on request from the corresponding author.

**Acknowledgments:** Authors wish to thanks State Forest of Tatra national Park, Šumava NP and to Military forests and farm of Czech Republic for providing data about tree mortality.

**Conflicts of Interest:** The authors declare no conflict of interest.

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
