# Peer review of "Wind Damage and Temperature Effect on Tree Mortality Caused by Ips typographus L.: Phase Transition Model"

_forests, doi:10.3390/f13020180_

Round 1
Reviewer 1 Report
Tree Mortality Caused by European Spruce Bark Beetle Ips typographus L. and Temperature: Phase Transition Model
Forests 1550976
This manuscript describes model development for understanding tree mortality. Overall, I think this manuscript is well written. I did notice missing spaces throughout (e.g., L22, L27, L112). In the results, the authors have place figures and text in awkward positions – one sentence paragraphs should be avoided and rewritten to provide better flow/structure for the reader.
Introduction
This provides a solid justification for the manuscript and I think the sources cited are appropriate.
Materials and Methods
L 95 – Check italics for names throughout.
L 95 – Should just cite Figure 1, don’t tell the reader to see it.
Figure 1 – Needs an elevation color definition. I would assume the darker reds are higher elevation, but a simple color ramp bar would be informative.
Table 1 – That many decimal places are necessary for the SA or MS coordinates, especially when the sites are 3,000-70,000 ha. Six decimal places is ~100 mm N/S and ~75 mm E/W at this latitude. I think two or three decimal places is plenty for the scale of this work.
Figure 2 – Line types need to be defined for the reader, even if it is in-text when referring to the figure, the reader needs guidance as to the function reaction to h>0, etc.
L 295, 315 – References equation 6, should this be 4?
L 306 – Capitalized However
Results
Figures 3-14 – The legends for defining the line types are not following any traditional standard for forestry or ecology journals. It would be easier for the reader if the authors would define the lines in the caption not using numerical values (solid, light dash, heavy dash, etc.).
L 378 – I’m not convinced R^2 is the best measure to use for selecting coefficient values. R^2 does not identify bias in the coefficients, is not a real measure of adequate fit, and is relative to that model providing little/no comparison to other models. I think more justification is needed as to the use of R^2 in this case (beyond the statement on L 313).
L 386-400 – Figure and text placement is extremely awkward and should be revised. L 388-389 and L393-395 are single sentence paragraphs. These need to be restructured to provide the reader with some flow/continuity in reading.
Figures 8-12 – I think axis limit consistency would make it easier for the reader to compare the site field and modelled values. Y-axis could easily be 0-14 and X-axis could be 1985-2015.
L 436-450 – Same comment as above – figure and text placement are awkward and need to be revised. Inappropriate to have single sentence paragraphs.
L 455 – This isn’t according to the tables – it is your interpretation of the values. Just make the direct statement “The order of autoregression for the model… equal to 2 (Tables 2-7).
Figure 14, Table 8 – Caption format error.
Discussion
The section is a little short, especially in interpreting the resulting model in context of the existing literature. There are a number of other pest outbreak models (not just bark beetle) that I think could be used to provide more context. Wildemeersch et al.(2019. Ecological Modelling 409:108745) used network models to predict outbreaks, Allstadt et al. (2013. Oecologia 172:141-151) looked at temporal shifts in cyclic outbreaks [seems the phase shift from your paper is modeling one wave of the cycle]. I think the authors could provide a better explanation of where their results fit in our overall understanding of outbreak occurrence within pest populations.
Conclusions
This should not reference tables. It should be a summary conclusion of meaning.
Author Response
Thank You for review. We understand all comments. We have tried to incorporate all suggestions into text.
We have just 2 points, where we were not able to follow reviewer suggestions:
1) Comment: „ Figures 3-14 – The legends for defining the line types are not following any traditional standard for forestry or ecology journals. It would be easier for the reader if the authors would define the lines in the caption not using numerical values (solid, light dash, heavy dash, etc.).“
Our response: The paper is on the margin of physics and forest ecology. We are using models used in physics to model bark beetle outbreak. The first author is physicist. We are using the style and graphics used in physics. We hope, the paper is interdisciplinary and the used style will help to general understanding.
2) Comment: “Figures 8-12 – I think axis limit consistency would make it easier for the reader to compare the site field and modelled values. Y-axis could easily be 0-14 and X-axis could be 1985-2015.”
Our response: We have considered modification, however It seems to us that changing the scale of the Y-axis will not improve the understanding of the model.
Reviewer 2 Report
My comments are in the attached word file.

Author Response
Thank You for review. We understand all comments. We have tried to incorporate all suggestions into text.
Reviewer 3 Report
The manuscript: „Tree Mortality Caused by European Spruce Bark Beetle Ips typographus L. and Temperature: Phase Transition Model” presents the development of a tree mortality model that explained the dynamics of trees (spruce) mortality caused by the bark beetle. The problem of modeling tree mortality caused by an outbreak of bark beetle is topical and an elaborated model can be relevant for the protection of unmanaged as well as managed spruce stands. In this form
General remarks
- Title of the manuscript – in this form temperature is equal to bark beetle – in the model-dependent variable (explained variable) is the amount of bark beetle (proxied by the amount of deadwood) temperature is explanatory variable influenced (or not in low elevation) on bark beetle population.
- A better description of data should be given especially basic characteristics of deadwood volume in study areas according to years– is it in form of time series?
- The theoretical background of the implemented model should be better described or relevant references should be cited – the general idea of the system in different states and transition between phases is sufficiently explained but the implementation of this theory to the particular case should be detailed described.
- The model consists of two submodels: long-term and short-term better description/explanation of how they are joined together should be given.
- The primary cause of tree mortality should be distinguished - as generalized in line 206 “Wind and beetle-caused wood damage (tree mortality) data were sourced from field surveys”
Specific comments
Line 33 ADL – consider using a full name instead of the abbreviation
Line 39 “Europe[1–10] insert a space Europe [1-10} – it concerns the whole manuscript: ( lines:47, 147, … )
Line 53. It should be “Duračiová et al. [39] …” in the same way source should be cited in line 246.
Line 128 „2.1.2.Š. umava” should be corrected.
Figure 2. Please add legend describing different lines
Line 295 “In Equation (6),” – probably concerns Equation 4, all symbols should be described.
equation 6 is given in line 380 in the form of first differences.
Line 322 – what kind of filter was used?
Line 353 Description of figure 3 should be improved.
Figure 5. In the description of the X-axis use “Lag k” instead of “Delay k”,
Line 371 It should be ‘“PACF” instead of ‘PFCF”
Line 391 “I suggest using “Empirical” instead of “Natural”
Author Response

(The authors gave the same response as above.)

Round 2
Reviewer 2 Report
The article is much improved from the first submission.